# Nonchemical Aquatic Weed Control Methods: Exploring the Efficacy of UV-C Radiation as a Novel Weed Control Tool

**DOI:** 10.3390/plants13081052

**Published:** 2024-04-09

**Authors:** Dian Udugamasuriyage, Gayan Kahandawa, Kushan U. Tennakoon

**Affiliations:** 1Centre for Smart Analytics, Institute of Innovation, Science and Sustainability, Federation University Australia, Gippsland Campus, Churchill, VIC 3842, Australia; d.udugamasuriyage@federation.edu.au (D.U.); g.appuhamillage@federation.edu.au (G.K.); 2Future Regions Research Centre, Institute of Innovation, Science and Sustainability, Federation University Australia, Berwick Campus, Berwick, VIC 3806, Australia

**Keywords:** UV-C, aquatic weeds, plant cell death, novel weed control methods, biological control

## Abstract

Aquatic weeds, including invasive species, are a worldwide problem. The presence of aquatic weeds poses several critical issues, such as hindering the continuous flow of water in irrigation channels and preventing the proper distribution of adequate water quantities. Therefore, effective control measures are vital for agriculture and numerous downstream industries. Numerous methods for controlling aquatic weeds have emerged over time, with herbicide application being a widely used established method of weed management, although it imposes significant environmental risks. Therefore, it is important to explore nonchemical alternative methods to control existing and emerging aquatic weeds, potentially posing fewer environmental hazards compared with conventional chemical methods. In this review, we focus on nonchemical methods, encompassing mechanical, physical, biological, and other alternative approaches. We primarily evaluated the different nonchemical control methods discussed in this review based on two main criteria: (1) efficiency in alleviating aquatic weed problems in location-specified scenarios and (2) impacts on the environment, as well as potential health and safety risks. We compared the nonchemical treatments with the UV-C-radiation-mediated aquatic weed control method, which is considered a potential novel technique. Since there is limited published literature available on the application of UV-C radiation used exclusively for aquatic weed control, our review is based on previous reports of UV-C radiation used to successfully control terrestrial weeds and algal populations. In order to compare the mechanisms involved with nonchemical weed control methods, we reviewed respective pathways leading to plant cell death, plant growth inhibition, and diminishing reemergence to justify the potential use of UV-C treatment in aquatic habitats as a viable novel source for aquatic weed control.

## 1. Introduction

The presence and growth of aquatic weeds in irrigation channels have imposed several challenges worldwide, especially related to the agricultural sector, which is dependent on irrigated water [1,2,3]. A continuous supply of irrigation water to the farm fields is essential in agriculture. Water is a primary resource used by the agricultural sector in the world, and since water has become a scarce resource, it is essential that water is used efficiently in our irrigation channels. For numerous countries, including Australia, efficient irrigation water management in farmlands has become a significant issue [4].

Negative effects of aquatic weeds not only affect irrigation channels but a wide variety of other natural and man-made aquatic habitats, like lakes [5,6,7], ponds [8], rivers [9,10,11,12], reservoirs [13], and coastal lagoons [14]. The spread of aquatic weeds can significantly impact various commercial activities. One of the primary economic concerns is their disruption of water transportation, directly affecting trade and tourism. Additionally, recreational activities such as swimming, boating, and fishing can be severely hampered by dense growths of aquatic weeds. Moreover, excessive weed growth can worsen flooding in waterways, posing risks to people’s safety, livelihoods, and environmental health. Another detrimental effect of aquatic weeds is their ability to degrade water quality, potentially leading to health hazards (i.e., eutrophication caused by increased growth of submerged aquatic weeds). Furthermore, irrigational activities suffer due to reduced water flow and damage inflicted upon irrigation equipment.

The use of chemical herbicides is the most common and popular method for controlling aquatic weeds worldwide. However, research evidence from Smalling et al. [15], Lee et al. [16], and Lopes et al. [17] highlights the undesired effects of chemical control methods in irrigation channels, posing a significant threat to aquatic organisms and public health. Dugdale et al. [18] identified three main chemicals used for weed management in Australia: Diquat (6,7-dihydrodipyrido (1,2-a:2’,1’-c) pyrazinediium dibromide), Acrolein (prop-2-enal), and Endothal acid (7-Oxabicyclo [2.2.1]heptane-2,3-dicarboxylic acid). One major drawback of this control method is its negative impact on other living organisms in aquatic habitats. Ortiz et al. [19], in their review, reported a major limitation of using herbicides in aquatic environments—the need for technical approvals due to the potential negative effects of the active chemicals on living organisms. They further suggest that integrated weed management can serve as an alternative to chemical control in aquatic weed management. Thus, this review will focus on various nonchemical aquatic weed control methods, including biological controls published from 1985 onwards to synthesize information on their effectiveness and environmental impacts. Further, suitability and the potential of UV (ultraviolet) radiation will also be assessed as a novel alternative aquatic weed control method.

## 2. Aquatic Weeds

Aquatic plants are plants that predominantly exist in water, either floating or submerged. These are considered aquatic weeds if a particular species causes problems, is considered undesirable to the natural aquatic habitats, and threatens native plants by out-competing them. One of the major issues that can arise when a certain aquatic plant has grown to more than the manageable level is the covering of the total water surface, leading to oxygen depletion underwater, which could adversely affect the underwater ecosystem [20]. A reduction in biodiversity due to the undesired growth of introduced aquatic plant species compared with native species is another negative effect, and this can further affect surrounding habitats and bird lives. Moreover, there are many other negative impacts on commercial activities due to the blockage of irrigation channels and waterways.

Aquatic weeds can be broadly classified into four categories [19]:(1)Emersed or emergent weeds growing on the banks of water or shallow water, with their stems and leaves found over the water’s surface. Examples of emergent aquatic weeds are Cattail (*Typha latifolia*) [21], Common rush (*Jucus usitatus*), and Water couch (*Paspalum distichum*).(2)Algae can be found floating on water surfaces with no identifiable structures of stems, leaves, or roots. Examples of algae are Filamentous green algae (*Cladophora* sp., *Spirogyra* sp.), Chara, and Nitella.(3)Floating-type weeds can also be found on water surfaces but with identifiable structures, and these can be rooted or nonrooted. Examples of floating aquatic weeds are Water hyacinth (*Eichhornia crassipes*), Salvinia (*Salvinia molesta*), and Water lettuce (*Pistia stratiotes*).(4)Submersed or submerged weeds are rooted in the bottom of the water and grow within the water body up to the surface. Examples of submerged aquatic weeds are Elodea (*Elodea canadensis*), Hydrilla (*Hydrilla verticillata*), and Ribbonweed (*Vallisneria australis*).

The presence of aquatic weeds has negative implications for agricultural irrigation, fishing, the quality of drinking water, habitats for fish and wildlife, flood control, human and animal health, hydropower generation, and the overall value of land [22]. Eiswerth et al. [23], in their study, evaluated the economic effects of Eurasian watermilfoil (*Myriophyllum spicatum*) in western Nevada and northeastern California, and they reported that even a 1% change in the aquatic weed vegetation level can significantly deteriorate economic outcomes of the recreational activities in lakes.

In all parts of the world, the general practices of managing aquatic weeds are categorized into four main areas: biological control, chemical control, mechanical control, and physical control [24,25]. However, there are some other methods, such as the drawdown technique, which are mainly used along with mechanical and physical control methods [5].

## 3. Mechanical Control of Aquatic Weeds

In a review, Zehnsdorf et al. [25] synthesized the mechanical methods that are used to manage aquatic weeds. The mechanical techniques summarized in the review include harvesting, cutting, dredging, and jute matting. They further mention that, in some cases, the application of the weed control method would increase the regrowth of the plants. Apart from that, the report mentions that the lowering of water levels in reservoirs or ponds is a common technique applied prior to mechanically removing these plants. This is called the drawdown technique. A comprehensive study was conducted in Argentina to assess the effectiveness of the chain-cutting technique on *Potamogeton illinoensis* Morong. The study evaluated how the application of the chain-cutting method affected the weed’s biomass weight [26]. Though the application of chain cutting has reduced the biomass weight of the weed, the researchers claimed that the use of this method increases the propagation of the weeds due to fragmentating plant parts, thus enabling viable rhizomes and stem fragments to move downstream [26,27]. This can be a major drawback faced by the mechanical control of weeds, unlike chemical control methods.

It is evident that there can be indirect advantages of aquatic weed harvesting. For example, Marwoto et al. [7] reported that mechanical harvesting of *Eichhornia crassipes* in the Indonesian lake, ‘Lake Rawa Pening’, led to resolving the issue of controlling invasive apple snails (*Pomacea canaliculata).* One of the main challenges of the mechanical removal of aquatic weeds can be the control of the collected weed biomass. However, there are many recent studies that focused on transforming this biomass into productive byproducts like compost fertilizer. Islam et al. [28] attempted to use water hyacinth as a raw material for producing handmade paper. Additionally, they utilized the generated waste to enhance the nutritional quality of compost. Weeds like *Hydrilla verticillata*, harvested mechanically, are combined with varying proportions of cow dung and sawdust to create stable compost. This compost has potential for utilization in agricultural systems [29].

## 4. Physical Control of Aquatic Weeds

Weed control using physical means is a widely used approach with habitat manipulation techniques [30], such as water level manipulation and reduction in photosynthetically active radiation availability. One commonly used physical control method is laying bottom benthic barriers to control the emergence of submerged weeds [31]. The materials used to make benthic bottom barriers can be both porous and nonporous, where the typical examples include jute, polyethylene [(C_2_H_4_) nH_2_], fiberglass fabric, and rubber. However, the applicability of this method varies according to the weed plant targeted to control [32] and the materials used to make the benthic barriers. Thus, it can be used to control weeds selectively while promoting the growth of native plants in aquatic systems [32]. To enhance the efficiency of weed control using benthic barriers, it is often integrated with chemicals. Barr and Ditomaso [31] demonstrated the effectiveness of acetic acid combined with benthic barriers in controlling curlyleaf pondweed (*Potamogeton crispus*) turion and followingly experimented with the successful control of the same weed using heated water [33]. In Ireland, promising results from pilot field trials demonstrated the efficacy of biodegradable jute material in eradicating the invasive aquatic macrophyte *Lagarosiphon major* from Lake Lough Corrib, as well as later restoring native macrophyte communities [34]. Another report mentioned combining chemicals (Dichlobenil (2,6-Dichlorobenzonitrile)), and mechanical ways (cutting) were unsuccessful in the long-term control of *Myriophyllum verticillatum* L. in Irish canals, but a physical method of turion removal in channels using a boat-mounted weed harvester was much more effective in managing the weed [35].

## 5. Biological Control of Aquatic Weeds

Biological control is another technique used for the control and management of aquatic weeds. A review conducted by Cuda et al. [36] outlined different biocontrol methods used in the USA to manage aquatic weeds, including *Hydrilla verticillata*, *Myriophyllum spicatum* (Watermilfoil) and *Egeria densa* using agents like anthropods, fish, and pathogens. This review further summarizes other factors, such as weather, habitat conditions, and biotic factors (host quality, matching genotypes), which affect the success or failure of biological weed control programs. Hence, it is evident that the success rate of a biocontrol program depends on numerous external factors [36,37]. Another review compiled by Pipalova [37] presented the effects of grass carps (*Ctenopharyngodon idella* Val.) on water bodies. The study reported that the aquatic macrophyte diversity can change due to the introduction of grass carp because the carp not only feed on the targeted aquatic weeds but on other plants too. Further, the review reported the changes in water quality parameters (physical and chemical) after the stocking of grass carp. A review compiled by Kumar [38] stated that grass carp (*Ctenopharyngodon idella*) was introduced to India from Japan to control aquatic weeds *Vallisneria* spp. and *Hydrilla verticillata.* Baars et al. [39] studied the possibility of controlling *Lagarosiphon major* (African curly-leaved waterweed) in Ireland using the natural bio enemies from Africa and further stated that though there are positive environmental implications in biological control, more research is needed in the area of host specificity. Research conducted in India on using the fungal pathogen *Alternaria alternata* to control the aquatic weed *Eichhornia crassipes* has shown significant influences of temperature, light, and incubation period on conidia production and virulence, suggesting potential for biological control in village cooperative scenarios [40]. Barbinta-Patrascu et al. [41] synthesized silver nanoparticles (AgNPs) utilizing aqueous extract from *Citrus reticulata* peels and studied the impact of these ‘green’ AgNPs on four invasive wetland plants, Cattail (*Typha latifolia*), Flowering-rush (*Butomus umbellatus*), Duckweed (*Lemna minor*), and Water-pepper (*Polygonum hydropiper*), revealing diverse responses across species and hinting at the potential efficacy of Citrus-based AgNPs as a biopesticide for pathogen and weed control in aquatic settings. As per a review by Ortiz et al. [19], biological control for managing aquatic weeds may be effective in certain specialized situations, but they are often challenging to execute on a broader scale. Moreover, these approaches may be less reliable or less predictable in achieving weed control outcomes compared with chemical weed management.

Table 1 offers a brief overview of a limited number of studies where biological control agents were employed for the purpose of aquatic weed management.

## 6. Other Techniques of Aquatic Weed Control

There are some other techniques that were used in several studies for aquatic weed control. In a study conducted by Abbot et al. [14], researchers examined the consequences of reinstating tidal flow in the coastal wetlands of the Boolgooroo lagoon area within the Mungalla wetlands, Australia. They investigated how this action, following the removal of earth bunds, affected aquatic weed infestation, water quality, and fish biodiversity. The results show that reintroducing tidal flow significantly reduced aquatic weed infestation, improved water quality, and tripled fish diversity. This suggests that reinstating tidal flow is a cost-effective and sustainable method for restoring wetlands.

The identification of weed vegetation in aquatic environments is equally important as controlling it, because it can lead to a more effective controlling program. A study conducted in the Guadiana River in Spain addresses the issue of aquatic invasive plant spread by developing a methodology for automatically detecting their geolocation using remote sensing and machine learning techniques [11]. The study introduces a methodology for validating classification results using synthetic ground truth images based on high spatial resolution imagery. Results indicate the effectiveness of certain algorithms in automatically detecting aquatic weeds by analyzing medium spatial resolution satellite images.

Due to safety issues and operational efficacies, there is a growing interest in unoccupied aerial application systems (UAASs) for herbicide applications. This novel technique can overcome issues with site access restrictions and environmental limitations, as well as improve ground-based applicator safety in various herbicide applications [56]. However, the application is currently suitable for only relatively small areas (<40,000 m^2^) and for sites where access from ground vehicles is impossible. A similar kind of study was conducted in Australia, where the researchers utilized Remotely Unmanned Aerial Vehicles (RUAVs) equipped with low-cost sensor suites and various weed detection algorithms to locate and identify weeds in inaccessible locations, particularly aquatic habitats. Additionally, the RUAVs were fitted with a spray mechanism to enable autonomous or remote-controlled spraying and treatment of the identified weeds. This system was successfully demonstrated in infested aquatic habitats where traditional access was difficult [57].

There are new studies where nonherbicide chemicals are evaluated for aquatic weed control. These are categorized as alternative chemical control of aquatic weeds. Acetic acid and d-Limonene were evaluated in Florida, USA for the control of several floating aquatic weeds, namely water hyacinth (*Eichhornia crassipes*), water lettuce (*Pistia stratiotes*), feathered mosquito fern (*Azolla pinnata*), common salvinia (*Salvinia minima*), rotala (*Rotala rotundifolia*), and crested floating heart (*Nymphoides cristata*). However, the researchers specifically mentioned in all the reports that the use of alternate chemical control is highly expensive compared with traditional herbicide applications, and it would be an option only when there are strict controls for herbicide applications on affected habitats [58,59,60].

## 7. Why UV-C as an Alternative/Novel Aquatic Weed Control Method?

The use of ultraviolet radiation for weed control was first evaluated by Andreasen et al. [61] for a few terrestrial weeds, such as annual bluegrass (*Poa annua* L.), common groundsel (*Senecio vulgaris* L.), shepherd’s purse (*Capsella bursa-pastoris* (L.) Medicus), small nettle (*Urtica urens* L.), canola (*Brassica napus* L. ssp. *napus*), and pea (*Pisum sativa* L.), and the authors proposed that further research was necessary to decide on the exact cellular mechanisms responses with this method and related health hazards due to overexposure of UV.

Though UV-C was tested as an alternative weed control method, the majority of terrestrial weed control methods currently used worldwide are manual/hand removal [62], mechanical removal [63,64], chemical control/herbicides [65], biological control [66,67], and fire [68]. Each control method possesses both pros and cons. One such risk of using ultraviolet radiation is the occupational health and safety hazard. However, unlike the common chemical control methods, UV-C application has an incidental effect, and it is vital to follow standard protocols during field applications [69].

There is one report and an associated patent for application technology (boat bottom mounted) for using UV-C radiation for aquatic weed control in Lake Tahoe, USA [70,71,72]. In this study, UV-C was used to control submersed aquatic weeds in Lake Tahoe, in 2018, in California, USA. The report states that using UV-C for invasive weed control was successful [72]. Another study evaluated the effects of UV-C radiation on *Juncus effusus*, a plant grown at the edges (banks) of aquatic habitats [73]. Further, there is abundant literature available on controlling algae using UV-C radiation [74,75,76,77,78,79,80,81]. Santos et al. [82] reported the effect of ultraviolet radiation on bacteria. When compared with the other two UV spectral regions (UV-A and UV-B), the lowest survival rate of bacteria was observed when exposed to UV-C radiation.

It is vital to study the possibility of using UV-C applications as weed management strategies, as well as their mechanisms. Many alternative aquatic weed control techniques were successful due to their process of inhibiting major biological and physical factors necessary for the growth and development of the weed plant, such as light for photosynthesis and/or nutrients for growth and development [6,34]. The UV-C region is defined as 100–280 nm wavelength [69]. Within this region, the 254 nm wavelength has been used in a majority of studies because of the commercial availability of the 254 nm UV-C sources [83,84,85,86] and due to the fact that the 254 nm wavelength is close to the peak germicidal wavelength of 262 nm [87]. The radiation energy can impact overall plant growth and thereby induce cell death. However, in the Tahoe weed control project conducted in 2019, only visual evidence of plant collapse resulting from the utilization of UV-C emitters was reported [72].

## 8. Ultraviolet Radiation Impact on Plant Cell Death

Ultraviolet radiation has three wavelength bands in its spectrum. These are UV-A (320–400 nm), UV-B (280–320 nm), and UV-C (200–280 nm). The intensity of the radiation or the irradiance is presented in W/m^2^ and the radiation dose is the multiplication of the irradiance and the exposure time, which is expressed in J/m^2^ [88].

Plant cell death can occur in two biological pathways, one is apoptosis and the other is necrosis. Apoptosis or programmed cell death is physiological cell death, whereas necrosis is accidental cell death. There are several differences in these two pathways, and they can be detected using experimental methods. These differences between apoptosis and necrosis with respect to plants were widely discussed by Danon et al. [89], Fomicheva et al. [90], Reape et al. [91], and Palavan-Unsal et al. [92] in their respective review articles.

As per a review compiled by Nawkar et al. [93], ultraviolet radiation affects DNA, proteins, lipids, and biochemical processes involved in photosynthesis. Further, this investigation reports that recent studies have discovered that an independently existing reactive oxygen species (ROS) is formed when plants are exposed to UV radiation, and some genes involved in the cell death pathway react to the radiation. The first evidence of plant cell death resulting from UV-C radiation was reported by Danon and Gallois [84] while studying UV-C radiation’s impact on *Arabidopsis thaliana.* In their study, *Arabidopsis thaliana* protoplasts were exposed to different doses of UV-C radiation. For radiation levels ranging from 10 kJ/m^2^ to 50 kJ/m^2^, DNA laddering was observed, indicating DNA fragmentation.

Another study [85] using *Arabidopsis thaliana* focused on levels of reactive oxygen species (ROS) and the behavior of mitochondria with the initial stage of plant cell death (PCD) induced due to UV-C exposure. A quick production of ROS was observed in the initial stages of PCD, and mitochondrial dysfunction was also observed in this investigation.

A few studies have reported visual and experimental evidence of the effects of UV-C at the whole-plant level. For example, Attia et al. [83] studied the effects of UV-C on lettuce plant leaves, and they observed necrotic spots in leaves when plants were exposed to higher UV-C doses (1.71 kJ/m^2^ and 3.42 kJ/m^2^). Further, this particular study shows the disruption of the photosynthetic machinery of the leaves. A study conducted by Najeeb et al. [73] using UV-C radiation to control mat rush (*Juncus effusus* L.) observed reductions in chlorophyll content and notable changes at the ultracellular level of the *Juncus effusus* plant leaves exposed to UV-C radiation. Some of these changes included a decrease in cell and chloroplast size, along with an increase in plastoglobuli within chloroplasts, leading to disruption of thylakoid integrity. This work suggests that UV-C radiation primarily damages chloroplasts, thereby subsequently triggering these modifications.

Figure 1 illustrates a summary of the aquatic weed control methods with their general mode of action and potential negative implications.

## 9. Conclusions and Prospects

Aquatic weeds pose a significant threat to the effective utilization of lakes, water channels, and irrigation systems. Chemical weed control, primarily through herbicides, remains the most common approach, despite their drawbacks related to adverse environmental impacts and high costs.

Various conventional methods for aquatic weed control exist, including mechanical, physical, and biological approaches, each having its own advantages and disadvantages. Mechanical and physical methods, while effective, are labor-intensive and may enable the regrowth of weeds due to stem fragments. Biological control methods, although environmentally preferable, are influenced by numerous external factors, and further research is needed to optimize their effectiveness.

The exploration of UV-C radiation as a novel method for aquatic weed control has shown promising results. Field studies, such as those conducted in Lake Tahoe, USA, have demonstrated its potential to control submerged and emergent aquatic weeds. Additionally, UV-C has been extensively utilized to control algae growth, although studies on its impact on plant health have so far focused on terrestrial weeds.

This review particularly aims to open a new direction toward integrating the use of UV-C treatment with other eco-friendly management actions such as bioherbicides [67] as a long-term solution over traditional synthetic herbicide applications. These integrated, nonchemical eco-friendly weed control practices will undoubtedly help reduce chemical pollution and the development of herbicide resistance in environmental weeds. It must be stressed that despite showing promising results as a novel weed control method, there are extensive gaps in research to focus on cellular changes in weeds caused by UV-C treatment and the viability of commercialization processes of UV-C applications when trying to implement weed control in extensive aquatic and terrestrial habitats.

The commercialization of UV-C applications can be achieved through tested technologies like autonomous underwater vehicles (AUVs) [94], suitable for aquatic environments. Additionally, UV-C applications can effectively target submerged weeds by installing radiation sources onto boat bottoms or floating equipment, particularly in lake environments. In irrigation channels, UV-C applications can be executed from the banks using mechanical arms. These applications hold the potential for further development and automation using emerging technologies. However, thorough research and assessment of commercial feasibility are necessary before implementation.

Given the evidence suggesting the potential efficacy of UV-C radiation in controlling aquatic weeds, further research is warranted to understand its genetic effects on plants and potential impacts on nontarget organisms. While negative effects may arise, they are typically incidental and manageable, indicating UV-C application as a commercially viable tool for controlling aquatic weeds in the future. For the present, we have reported the most recent research findings in this review. Undoubtedly, a plethora of complex questions will continue to emerge relating to the overall impacts of some of these novel weed control methods on the biophysical environment. Well-coordinated, interdisciplinary research efforts are called for to find answers and definitive solutions for such issues. 

## Figures and Tables

**Figure 1 plants-13-01052-f001:**
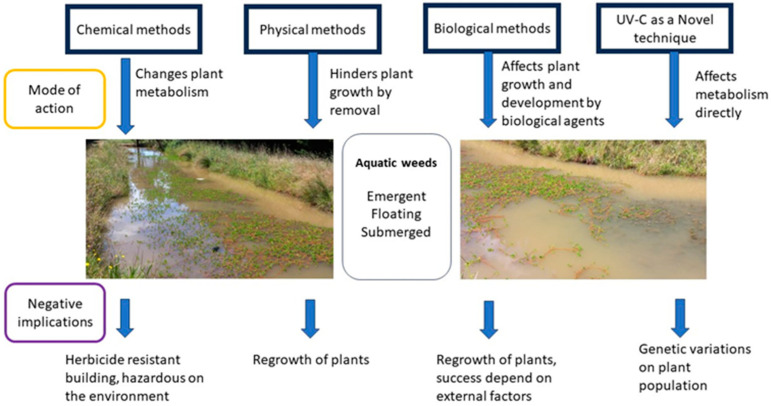
Schematic diagram showing the general modes of action in controlling different aquatic weeds and potential negative implications of each method (Dian Udugamasuriyage).

**Table 1 plants-13-01052-t001:** Different biocontrol agents used to control aquatic weeds.

Life Form of Selected Aquatic Weeds	Scientific Name of Weed	Control Mechanism	Control Agent	Reference
Floating	*Eichhornia crassipes*	Fungal infection effect on plant growth	*Diaporthe eres (Phomopsis oblonga)*	[42]
Submerged	*Hydrilla verticillata*	Leaf damage by larvae	*Hydrellia pakistanae*	[43]
Submerged	*Lagarosiphon major*	larvae feed on actively growing shoots	*Polypedilum tuburcinatum*	[44]
Floating	*Eichhornia crassipes*	Fungal pathogen	*Alternaria alternata*	[40]
Floating	*Salvinia molesta*	Weevil feeding on leaves	*Cyrtobagous salviniae*	[45]
Submerged	*Hydrilla verticillata*	Leaf damage by larvae	*Hydrellia balciunas*	[46]
Submerged	*Hydrilla verticillata*	Feeding on plant	*Bagous hydrillae*	[47]
Submerged	*Hydrilla verticillata*	Larvae feeding on growing tips	*Cricotopus lebetis*	[48]
Floating	*Eichhornia crassipes*	Feeding on plant	*Neochetina eichhorniae*	[49]
Submerged	*Hydrilla verticillata*	Feed on plant	*Parapoynx diminutalis*	[50]
Floating	*Eichhornia crassipes*	Feed on leaves	*Cornops aquaticum*	[51]
Algae	*Cladophora globulina*	Feed on algae	*Ctenopharyngodon idella*	[52]
Algae	*Phaeocystis globosa*	Release of algicidal substances	*Bacillus* sp.	[53]
Emergent	*Lythrum salicaria*	Feed on plant	*Hylobius transversovittatus*	[54,55]

## Data Availability

Not applicable.

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
