# Peer review of "Nonchemical Aquatic Weed Control Methods: Exploring the Efficacy of UV-C Radiation as a Novel Weed Control Tool"

_plants, 2024, doi:10.3390/plants13081052_

Round 1

Reviewer 1 Report

Comments and Suggestions for Authors

General comments:

This paper is a well-written mini-review on techniques for aquatic weed control beyond that of herbicides. The review quickly recaps some of the traditional non-chemical control mechanisms (such as mechanical or physical control) and then turns the focus towards a potential emerging control technique of UV radiation. Overall, the review is well-written and clear. The body of information available for UV-C as a weed control method is limited, but the authors do call out the need to investigate this further. With the exception of checking for minor editorial corrections, this review paper is suitable for publication.

Specific comments:

No specific concerns.

Reviewer 2 Report

Comments and Suggestions for Authors

The manuscript entitled "Non-Chemical Aquatic Weed Control Methods: Exploring the Efficacy, with a Special Focus on UV-C Radiation as a Novel, Cost Effective Alternate Technique" is not new to the literature review and I do not recommend publication in Plants. See comments below.

Introduction: the name of the herbicides should be the active ingredient, only diquat is correct.

The introduction is very simple and does not show the problematics and innovation of the study.

Aquatic weeds: does not exemplify the classification of plants, the text is poor.

Manual control is mechanical and not physical.

Reviewer 3 Report

Comments and Suggestions for Authors

1 The title is too long.

2 Might the contents of introduced aquatic plan speciesin line 65 be introduced aquatic plant species or invasive alien aquatic plant species (IAAPs)? Please check it.

3 The numbers of 1-4 in lines 70-76 might be changed to (1)-(4).

4 Could you add the related contents of aglae in Table 1 which is one of four aquatic weeds mentioned in section " 2. Aquatic weeds" in lines 70-77?

5 Can you use non-chemicals directly instead of the expression of non-herbicide chemicals?

6 The 2 in m2 of line 272 should be in superscript.

7 We could not find the specific suggestions for the future researches on UV-C radiation as a novel method for aquatic weed control, so I think that the section 9. Conclusions could be changed to 9. Conclusions and Prospects and add some suggestions or prospects for readers that specific measures and a little bit more detailed research topics of UV-C radiation could be studied.

Round 2

Reviewer 2 Report

Comments and Suggestions for Authors

The corrections made are not sufficient for publication in Plants,